# Comparative Characterization and Immunomodulatory Activities of Polysaccharides Extracted from the Radix of *Platycodon grandiflorum* with Different Extraction Methods

**DOI:** 10.3390/molecules27154759

**Published:** 2022-07-25

**Authors:** Wanwan Xiao, Pingfan Zhou, Xiaoshuang Wang, Ruizhi Zhao, Yan Wang

**Affiliations:** 1School of Chinese Materia Medica, Guangdong Pharmaceutical University, Guangzhou 510006, China; w989700@163.com (W.X.); pinkgiee@163.com (P.Z.); 2The Second Affiliated Hospital of Guangzhou University of Chinese Medicine, Guangzhou 510006, China; m18314484252@163.com; 3Guangdong Provincial Key Laboratory of Clinical Research on Traditional Chinese Medicine Syndrome, Guangzhou University of Chinese Medicine, Guangzhou 510006, China

**Keywords:** polysaccharide, *Platycodon grandiflorum*, different extraction method, immunity

## Abstract

*Platycodon grandiflorum* is an edible and medicinal plant, and polysaccharides are one of its important components. To further improve the utilization rate of *P. grandiflorum*, we investigated the effects of four different extraction methods, including hot water, ultrasonic-assisted, acid-assisted, and alkali-assisted extractions, on the polysaccharides, which were named PG-H, PG-U, PG-C, and PG-A. The findings indicated that the extraction method had a significant impact on the yield, characteristics, and immunoregulatory activity. We observed that the yields decreased in the following order: PG-H, PG-U, PG-C, and PG-A. Galacturonic acid, glucose, galactose, and arabinose were the most prevalent monosaccharides in the four PGs. However, their proportions varied. In addition, the difference between the content of glucose and galacturonic acid was more significant. PG-U had the highest glucose content, whereas PG-C had the lowest. Galacturonic acid content was highest in PG-A, while the lowest in PG-U. The molecular weight decreased in the order of PG-U, PG-H, PG-C, and PG-A; the particle size was in the order of PG-U, PG-A, PG-H, and PG-C. Moreover, the extraction method had a great impact on immunoregulatory activity. The ability to stimulate the immune function of macrophages was as follows: PG-A > PG-C > PG-U > PG-H. The results indicated that PGs, with lower molecular weights and higher GalA content, exhibited better immune-stimulating activity. And more important the AAE method was a good way to extract polysaccharides from *Platycodon grandiflorum* for use as a functional product and immunological adjuvant.

## 1. Introduction

*Platycodonis Radix*, the rhizomes of *Platycodon grandiflorus* (Jacq.) A. DC., is widely distributed in Southeast Asia, including China, North Korea, Japan, and Mongolia. As an edible and medicinal herb, *P. grandiflorum* has beneficial effects against cough, excessive phlegm, and sore throat. Furthermore, it possesses antitumor, immunomodulatory, and antioxidative properties [1]. *P. grandiflorum* contains many chemical components, such as saponins, flavonoids, anthocyanins, phenolics, and polysaccharides [2]. Polysaccharides from *Platycodonis Radix* (PGs) are one of the essential active ingredients of *P. grandiflorum*, and have attracted broad attention due to their antitumor [3], antioxidative [4], and immunoregulatory effects [5]. Among the many properties of PGs, their immunomodulatory activity has increasingly gained attention [6]. Because of weight-loss, antibiotic abuse, environmental issues, and other factors, immunodeficiency has increased in recent years [7].

Macrophages are important immune cells that play a role in both innate and adaptive immunity, and the binding of polysaccharides to specific membrane receptors on cells is one of the most important ways to activate the immune response [8]. Activated macrophages can kill many pathogenic microorganisms, as well as engulf apoptosis-damaged and tumor cells. Additionally, they stimulate other immune cells to respond to pathogens. Previous studies showed that polysaccharides regulate macrophages in a variety of ways, affecting their number, morphology, phagocytic capacity, and cell secretion capacity [9].

Recently, due to a large number of studies on immunomodulation by natural products, their utilization rate has shown a rising trend [10,11]. The question of how to make better use of natural products is currently a popular topic. Extraction is the first and crucial step for the efficient utilization of natural products [12]. Many extraction methods are used in the process of polysaccharide extraction, including hot water extraction (HWE), ultrasonic-assisted extraction (UAE), acid-assisted extraction (CAE), and alkali-assisted extraction (AAE) [13]. However, to the best of our knowledge, the CAE and AAE methods have not been applied to extract polysaccharides from *P. grandiflorum*. Moreover, different extraction methods affect the yield of polysaccharides, and may result in variations in their structure and activity [14]. It is unknown which extraction method produces PGs with the highest yield and most robust immunoregulatory activity. There is also limited information about the correlation between the extraction methods and the characteristics and bioactivities of PGs. Therefore, in order to better develop PGs as a functional product and immunological adjuvant, the effect on activity and physicochemical characteristics of the different extraction methods on the PGs were compared in this study.

## 2. Results and Discussion

### 2.1. Extraction Yield and Carbohydrate Content of PGs

The results are listed in Table 1. The yield of PG-H was the highest (16.62%), followed by those of PG-U, PG-C, and PG-A. Compared with hot water extraction, alkaline extraction and acid extraction produced a lower yield. The reason behind this could be the hydrolysis of polysaccharides during alkaline and acid extraction into small saccharides that could not be ethanol-precipitated [15,16]. To prevent the hydrolysis of polysaccharides, it is essential to control both the alkaline and acid concentrations, as well as the extraction time during alkaline and acid extraction [16]. Among the four PGs, the carbohydrate content of PG-H was the lowest.

### 2.2. Characteristics of PGs

#### 2.2.1. FT-IR and UV Spectroscopy Analyses of PGs

The infrared spectra of the PGs are shown in Figure 1A. The broad and intense bands around 3333 cm^−^^1^ suggested the presence of O-H stretching vibrations, and the peaks near 2935 and 1415 cm^−^^1^ were attributed to C-H stretching and C-H bending, respectively [17]. Moreover, the noticeable adsorptions around 1730 cm^−^^1^ were consistent with C=O stretching vibrations, indicating that the PGs contained uronic acid. It is also worth noting that the typical protein bands at 1651 and 1555 cm^−^^1^ were not detected, suggesting that none of the PGs contained proteins [18]. The peaks near 1024 cm^−^^1^ were due to C-O-C stretching vibrations [19]. The absorption at 820 cm^−^^1^ suggested the presence of α-configurations [20]. The FT-IR spectra of the four PGs displayed similar characteristics, indicating that the different extraction methods did not influence the type of glycosidic bonds and polysaccharide conformations [21].

As shown in Figure 1B, the UV–vis spectra of the PGs revealed that there were no absorption peaks between 260 and 280 nm, suggesting that the PGs extracted by the four methods did not contain proteins [22]. This result was identical to that observed in the FT-IR spectroscopy.

#### 2.2.2. Molecular Weights of PGs

As shown in Figure 2 and Table 2, the average molecular weight (Mw) distribution of the four PGs mainly included two fractions: fraction I, which corresponded to the peak of polysaccharides with high molecular weight, and fraction II, which corresponded to the peak of polysaccharides with lower molecular weight. The separation of the polysaccharide mixture produced two main fractions, but we observed that the first fraction mainly consisted of two different polysaccharide peaks. In addition, the second fraction of PG-C consisted of the two different polysaccharide peaks.

Compared with PG-U, the molecular weights of PG-H, PG-C, and PG-A were lower at both peaks I and II. PG-A had the smallest molecular weight for both peaks I and II compared with the other three PGs. Moreover, for fractions of high molecular weight, the molecular weight was in the following order: PG-U > PG-H > PG-C > PG-A. This may have been due to the fact that acid and alkaline are more easily able to break the glycosidic linkages [23]. The molecular weight of PG-U exceeded that of PG-H; although ultrasound induced glycosidic linkages [24], the longer time at higher temperatures, by hot water extraction, also induced the degradation of polysaccharides [25]. For fractions of low molecular weight, the molecular weight of PG-H was similar to that of PG-U, and the molecular weight of PG-C was the same as that of PG-A.

#### 2.2.3. Monosaccharide Composition of PGs

The monosaccharide compositions of the PGs extracted by the four methods were analyzed by HPLC. As shown in Table 3 and Figure 3, the monosaccharide compositions of all four PGs were the same, and they were composed of mannose (Man), rhamnose (Rha), galacturonic acid (GalA), glucose (Glc), galactose (Gal), and arabinose (Ara). However, each PG had a different molar ratio. The molar ratio was 1.9: 2.7: 5.3: 22.8: 17.1: 50.1 in PG-H, 3.6: 3.2: 3.4: 42.2: 17.0: 30.6 in PG-U, 0.9: 4.2: 7.4: 5.8: 22.5: 59.2 in PG-C, and 2.2: 7.2: 23.9: 11.2: 15.1: 40.4 in PG-A. With the exception of PG-U, Ara had the highest percentages among the other three PGs: 50.1% (PG-H), 59.2% (PG-C), and 40.4% (PG-A). The percentage of Glc in PG-U (42.2%) was higher than in PG-H (22.8%), whereas the Glc percentage was lower in PG-C (5.8%) than in PG-H. Furthermore, PG-A had the highest GalA content, and the order was as follows: PG-A > PG-C > PG-H > PG-U.

#### 2.2.4. Particle Size and Zeta-Potential of PGs

As shown in Figure 4A, the particle size distribution of PGs mainly focused between the range of 200 and 400 nm. When compared with PG-U, PG-C, and PG-A, the particle size distribution of PG-H was broader. The particle size of PG-C was smaller than that of the other three PGs. The order of the particle sizes was similar to the molecular weights, but not identical. This could be explained by the fact that, in addition to molecular weight, the particle size of the polymer is associated with the degree of molecular chain extension and the aggregation state [15]. However, the particle size of PG-A was bigger, which may be caused by the crosslinking of aggregated polysaccharides containing higher Gal-A [26].

The zeta-potentials of the PGs are shown in Figure 4B. All PGs were negatively charged, but the zeta-potential of PG-A was the highest (−28.6 ± 3.3), followed by those of PG-U (−24.4 ± 2.8), PG-H (22.8 ± 1.5), and PG-C (−15.5 ± 2.8). All four PGs carried negative charges, owing to the existence of GalA [27]. This was in accordance with the results of the monosaccharide composition and FT-IR spectroscopy analyses. In addition, the more anion groups in the PGs, the better the stability of the PG-based solutions or colloids [28]. Accordingly, the stability of PG-A, with the most anion groups, was the best, followed by PG-U, PG-H, and PG-C.

### 2.3. Immunomodulatory Effects of PGs

#### 2.3.1. Effect of PGs on Macrophage Cells Proliferation

None of the PGs showed any effect on the macrophage’s proliferation in the range of 4 to 100 µg/mL.

#### 2.3.2. Effect of PGs on NO Production in Macrophage Cells

Activated macrophages can produce a large amount of NO and various cytokines, such as TNF-α and IL-6, which are involved in killing pathogens, microorganisms, and mediating various biological functions [29,30]. As depicted in Figure 5A, all four PGs promoted NO production in macrophages compared with the negative control. However, the NO-stimulating capacity of the PGs varied with different extraction methods. Compared with the control, PG-H at 100 µg/mL concentration had no effect on NO production, while PG-U at 100 µg/mL significantly increased NO production. In the case of PG-C and PG-A, NO production in macrophage cells prominently increased at 4 µg/mL.

#### 2.3.3. Effects of PGs on the Expression of TNF-α and IL-6 in Macrophage Cells

As shown in Figure 5B, all PGs enhanced the production of IL-6 in a dose-dependent manner at the determined concentration. We observed that the IL-6-stimulating ability of all four PG samples dropped in the order of PG-A, PG-C, PG-U, and PG-H. Compared with the control, the expression of IL-6 in macrophages treated with 4 µg/mL of PG-H, PG-U, PG-C, and PG-A was increased by about 0.16, 0.7, 35, and 40 times, respectively; however, at the concentration of 100 µg/mL, IL-6 was increased by about 6.5, 30, 66, and 106 times, respectively.

The effect of PGs on TNF-α expression was similar to that of IL-6 and is depicted in Figure 5C. Compared with the blank control, the expression level of TNF-α in macrophages treated with 4 µg/mL of PG-H, PG-U, PG-C, and PG-A was increased by about 0.5, 1.2, 4.7, and 2.4 times, respectively. Nonetheless, as the concentration of PGs increased, their effect on TNF-α changed. When macrophages were treated with PG-A at 100 µg/mL, TNF-α production was significantly higher than that of PG-C (*p* < 0.01), indicating that PG-A causes a significant effect on TNF-α production when used at higher concentrations.

#### 2.3.4. Effect of PGs on Phagocytosis in Macrophage Cells

Macrophages, as the most critical phagocytes, can eliminate exogenous pathogenic microorganisms and endogenous dead cells [31,32]. As shown in Figure 5D, the effect of PGs on phagocytosis was weaker than their effects on NO, IL-6, and TNF-α. With the exception of PG-C, the other three PGs slightly stimulated the phagocytosis of macrophage cells at 100 µg/mL.

Analyses of the immunomodulatory effects of PGs suggested that PGs enhanced the immune function of macrophages in the order of PG-A > PG-C > PG-U > PG-H, which may be associated with the ratio of monosaccharide composition, and different proportions of reducing and nonreducing ends. Higher galacturonic acid content induced stronger immunoregulatory activity, and this may be part of the reason for the strongest immunoregulatory effects of PG-A [33]. However, compared with PG-H, PG-U, with a higher molecular weight and lower galacturonic content, also exhibited better immunoregulatory activity.

All the PGs extracted by the four different methods could induce macrophages to regulate immune responses, but the effect of PG-A was significant, which is instructive for the development and utilization of PGs. Moreover, alkali-assisted extraction (AAE) was used for the first time to extract polysaccharides from *Platycodon grandiflorum*, which may also be a suitable method of polysaccharide extraction from other plants. Although the observations confirmed that PG-A had the best ability to enhance the immunological function of macrophages, unfortunately, it had a lower yield. To obtain PGs with better immunomodulatory activity and higher yields, extraction conditions ought to be optimized. Furthermore, it is unclear which one of the homogeneous polysaccharides exerted the most crucial influence. Future studies may include the determination of the active homogenous polysaccharides, and gaining insights into the structure–activity relationship by comparing homogeneous polysaccharides from PG-A.

## 3. Materials and Method

### 3.1. Materials

*Platycodon grandiflorus* (Jacq.) A. DC. (Lot No.: 201000029) was purchased from Kangmei Co., Ltd., (Guangdong, China) and was authenticated by Nengfeng Ou, the pharmacist in charge of herb quality in the Second Affiliated Hospital of Guangzhou University of Chinese Medicine (Guangzhou, China). Dextran standards: Dextran 4 K (Lot No.: 8842), Dextran 60 K (Lot No.: D620D1–9), Dextran 450 K (Lot No.: D620DC–3), Dextran 680 K (Lot No.: D620D8–2), Dextran 2370 K (Lot No.: D620D7–1), and Dextran 3690 K (Lot No.: D620D10–1) were purchased from American Polymers Standards Corporation (Cleveland, OH, USA). RAW264.7 cells were procured from the China Center for Type Culture Collection (Wuhan, China). DMEM medium (Lot No.: 8120498), fetal bovine serum (FBS) (Lot No.: 2176398), and penicillin–streptomycin (Lot No.: 2199829) were obtained from GIBCO (Gaithersburg, MD, USA). 3-(4,5-dimethylthiazolyl-2)-2,5-diphenyl tetrazolium bromide (MTT) (Lot No.: 7101KMP) was purchased from MP Biomedicals, LLC (Santa Ana, CA, USA,). Lipopolysaccharide (Lot No.: 2199829) was purchased from Jingxin Biological Technology Co., Ltd. (Guangzhou, China). Neutral Red (Lot No.: A0425A) was purchased from Dalian Meilun Technology Co., Ltd. (Dalian, China). Total Nitric Oxide Assay Kit for Griess (NO) (Lot: 022421210729) was purchased from Beyotime Biotechnology (Haimen, China). Enzyme-linked immunosorbent assay (ELISA) kits, for tumor necrosis factor-alpha (TNF-α) (Lot No.: EGJSPRYYDL) and interleukin-6 (IL-6) (Lot: XR9FHB6EYS), were procured from Wuhan Sci-meds Biopharmaceutical Co., Ltd (Wuhan, China).

### 3.2. Extraction of Polysaccharides

Before extraction, *Platycodon grandiflorum* (Figure 6) pieces were crushed and passed through 60-mesh sieves and then defatted four times with 70% ethanol (*v*/*w* = 5:1) for 2 h. Then, the sample was dried in a fume hood. After that, 100 g of the sample was extracted with 2500 mL ultrapure water [34], citric acid solution of pH 3.0 [35], and NaOH solution of pH 12.0 [35] for 3 h to obtain PG-H, PG-C, and PG-A, respectively. For PG-U, the same weight of sample and amount of ultrapure water were used, and 200 W ultrasonic waves at 70 °C for 25 min were applied twice [36]. Next, the extracted solutions were mixed and centrifuged. After centrifugation, NaOH aqueous was used to neutralize the supernatants extracted by citric acid, while the supernatants extracted by hot water, ultrasound, and NaOH were neutral. The supernatants were concentrated to 0.2 g/mL of crude extract content by rotary evaporator at 60 °C under vacuum conditions. Afterward, the concentrates were dialyzed for three days. Next, they were precipitated with ethanol until reaching a final concentration of 80% (*v*/*v*) and stored overnight at 4 °C. Later, the precipitates were washed three times with ethanol and then vacuum-dried. Finally, the extracts were dissolved in distilled water and deproteinized with Sevag reagent (chloroform and *n*-butanol, 4:1, *v*/*v*), and the water layer was vaporized using a rotary evaporator at 60 °C. Then, the sample was lyophilized to obtain PGs.

### 3.3. Determination of Extraction Yield

The extraction yield (%) of polysaccharides is listed in Table 1.

### 3.4. Determination of Carbohydrate Contents

The total carbohydrate content of PGs was determined by the phenol-sulfuric method [37]. d-glucose (10 mg), used as a standard, was dissolved in 10 mL of ultrapure water to prepare a 1 mg/mL stock solution, which was then diluted to standard solutions of different concentrations (500, 250, 125, 62.5, 31.25, 15.62, 7.81, and 3.90 µg/mL). Similarly, 4 mg of PGs was dissolved in 4 mL of ultrapure water to prepare a 1 mg/mL polysaccharide solution, and then polysaccharide solutions were diluted to 500 µg/mL and 250 µg/mL. The 500 µg/mL solution of PGs was utilized as a correction concentration [38]. Then, in light of the protocol for the phenol-sulfuric method, the total sugar content of PGs was determined.

The total carbohydrate content of PGs was calculated according to the standard curve and the formula as follows:(1)f=W/C1×D1
(2)Carbohydrate contents (%)=(C2×D2×f)/W
where *W* is the PGs’ quality; *C1* and *C2* are the concentrations measured when PGs were prepared at 500 and 250 g/mL, respectively; and *D1* and *D2* are the diluted multiples when PGs were prepared at 500 µg/mL and 250 µg/mL, respectively.

### 3.5. Structural Characteristics of PGs

#### 3.5.1. Fourier Transform Infrared (FT-IR) Analysis

FT-IR spectra of polysaccharides were obtained by an FT-IR spectrophotometer (Spectrum Two, PerkinElmer, Waltham, MA, USA) within the frequency range of 4000–400 cm^−1^ [39].

#### 3.5.2. UV Analysis

UV–vis spectra of 1 mg/mL polysaccharide solutions were measured using a UV–vis spectrophotometer (U-2910 ultraviolet spectrophotometer Hitachi Hi-Tech Co., Tokyo, Japan) within the wavelength range of 190–400 nm [37].

#### 3.5.3. Molecular Weight Determination

The molecular weight distribution of the PGs was determined using the high-performance gel permeation chromatography (HPGPC) method [40]. The sample solutions (1 mg/mL), prepared in ultrapure water, were first filtered through a 0.45 µm filter, and then the injection volume of 10 µL was loaded onto a TSK gel G5000PWXL column (7.8 mm × 300 mm, Tosoh Corporation, Yamaguchi, Japan). The column was maintained at 45 °C, and the elution was conducted at a flow rate of 0.8 mL/min. A series of dextran standards with different molecular weights (Mw) (4, 60, 450, 680, 2670, and 3690 kDa) were used to obtain the calibration curve, which was then used to calculate the molecular weights of the PG samples.

#### 3.5.4. Monosaccharide Composition Determination

Based on the method of Zhao et al., the monosaccharide composition was measured [41]. We hydrolyzed 5 mg of the PG samples, extracted using the four methods with 5 mL of 4 mol/L trifluoroacetic acid (TFA, CF_3_COOH) solution at 100 °C for 4 h; excess TFA was evaporated with ethanol after the hydrolysis. After that, 200 µL of the hydrolyzed samples was mixed with 200 µL of 0.6 mol/L 1-phenyl-3-methyl-5-pyrazolone (PMP) and 200 µL of 0.3 mol/L NaOH solution, and incubated at 70 °C for 100 min. After cooling to room temperature, 200 µL of 0.3 mol/L HCl solution was used for neutralization of the resultant products. Then, 0.6 mL of chloroform was added for extraction and centrifuged to obtain the sample solutions. These sample solutions were filtered through a 0.45 µm membrane and then analyzed by Agilent 1260 HPLC (Agilent Technologies, Santa Clara, CA, USA) with diode array detector (DAD) at a wavelength of 254 nm, and with a Diamonsil C18 column (250 × 4.6 mm, 5 µm Dikema Technology Co., Ltd., Beijing, China). The mobile phase for elution was 0.05 mol/L phosphate buffer (Na_2_HPO4-NaH_2_PO_4_, pH = 6.83) plus acetonitrile (83:17, *v*/*v*). The flow rate was 1 mL/min, the column temperature was 40 °C, and all sample solutions were analyzed at 254 nm. Standard monosaccharide solutions were prepared and analyzed as mentioned above. The monosaccharide compositions of the PGs were affirmed by each standard monosaccharide’s retention time. Then, standards, which were the same in monosaccharide composition as the PGs, were prepared at different concentrations of mixing solutions (500, 250, 125, 62.5, 31.25, and 15.625 µg/mL) to be derived and measured. Then, the ratio of monosaccharides of the four PGs was determined from the peak area of mixing solution.

#### 3.5.5. Molecular Particle Size and Zeta-Potential Analysis

Particle size distribution and zeta-potential of the PG solutions (1 mg/mL) were measured using a NanoBrook 90Plus PALS instrument (Brookhaven Instruments, Holtsville, NY, USA) and analyzed by Particle Solutions v.3.6.07122 (Holtsville, NY, USA).

### 3.6. Measurement of Immunomodulatory Activity

#### 3.6.1. Macrophage Proliferation Assay

RAW264.7 cells were seeded in 96-well plates with a density of 3 × 10^4^ cells/mL and incubated for 24 h. Then, the culture medium was removed, and 200 µL of PG-H, PG-U, PG-C, and PG-A extracts at different concentrations (100, 20, 4 µg/mL), respectively, was applied to the cells and incubated for 24 h. The culture medium served as a negative control. At 24 h post incubation, 20 µL of MTT (5 mg/mL) was added to each well, and the cells were re-incubated for 4 h. Then, the supernatants from each well were replaced with 150 µL of DMSO, and the absorbance was measured by a microplate reader at 490 nm.

#### 3.6.2. Measurement of NO

RAW264.7 cells were seeded at a density of 3 × 10^5^ cells/mL in 24-well plates and incubated for 24 h. Then, the culture medium was removed, and the cells were treated with 1 mL of all four PG solutions at different concentrations (100, 20, or 4 µg/mL). The culture medium served as a negative control, and 1 µg/mL lipopolysaccharide (LPS) was added as a positive control. After treatment for 24 h, the supernatants were collected to determine NO concentrations using a Griess kit [42].

#### 3.6.3. Measurements of TNF-α and IL-6

RAW264.7 cells were seeded in 24-well plates (5 × 10^5^ cells/mL) and incubated for 24 h. Similarly, cells were treated as mentioned in Section 3.6.2 and then the supernatants were collected for analysis by the ELISA kits as per the manufacturer’s instructions [43].

#### 3.6.4. Phagocytosis Assay for Macrophages

The phagocytic potential of macrophages was determined by the neutral red uptake method [44]. Firstly, RAW264.7 cells were seeded in 96-well plates (4 × 10^4^ cells/mL) and cultured for 24 h. We added 1 mL of the PGs to cells at different concentrations (100, 20, or 4 µg/mL). The culture medium was used as a negative control, and 1 µg/mL LPS was used as a positive control. After 24 h of incubation, the treatment solutions were discarded, and the cells were washed once in PBS and then treated with 100 µL of neutral red solution (1 µg/mL) for 30 min. After that, the supernatants were removed, and the cells were washed thrice with PBS. We added 100 µL of cell lysate (ethanol and 1.0 mol/L acetic acid at the ratio of 1:1, *v*/*v*) to cells and kept overnight to lyse the cells. Finally, the optical density for all PGs was measured using a microplate reader at 540 nm.

### 3.7. Statistical Analysis

The data are expressed as the mean ± standard deviation (S.D.). Origin 2021 (OriginLab Corporation, Northampton, NC, USA) and GraphPad Prism 8 (GraphPad Software Inc., San Diego, CA, USA) were used for analysis and graphing. The significance of the difference was evaluated with analysis of variance (ANOVA), followed by Tukey’s test using GraphPad Prism 8; *p*-values < 0.05, 0.01, or 0.001 were considered statistically significant.

## 4. Conclusions

Polysaccharides extracted using various methods can be different in their structure and activity [45]. In this study, PGs were extracted by different methods and their characteristics and immunomodulating activities were assessed. The results showed that the extraction method had significant effects on the yield; and characteristics such as particle size, molecular weight, the ratio of monosaccharide composition, and the immunoregulatory activity of PGs, but it had little impact on FT-IR and UV–vis spectra. It was also found that, although the traditional water extraction method could obtain the highest yield, the ability of this extract to enhance the immune activity of macrophages was far lower than the other three PGs. Meanwhile, PG-A exhibited the strongest immunomodulating activity. Therefore, alkali-assisted extraction is a better extraction method to produce PGs with greater activity.

## Figures and Tables

**Figure 1 molecules-27-04759-f001:**
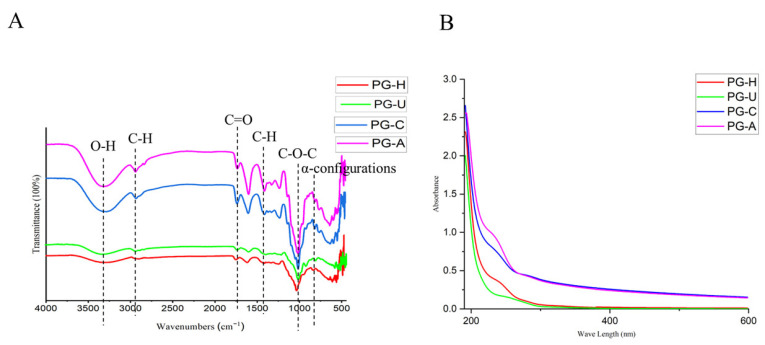
(**A**): FT-IR signals of PGs obtained by different extraction methods. (**B**): UV–vis spectra of PGs obtained by different extraction methods.

**Figure 2 molecules-27-04759-f002:**
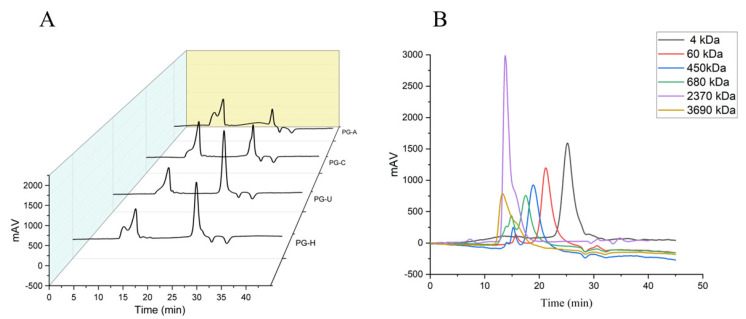
(**A**) Molecular weights of PGs obtained by different extraction methods. (**B**) Molecular weight of the dextran standards.

**Figure 3 molecules-27-04759-f003:**
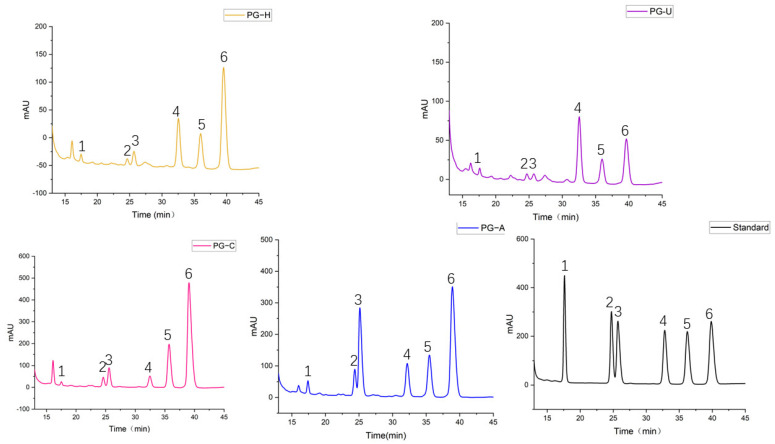
Monosaccharide composition of PGs obtained by different extraction methods. 1: Man, 2: Rha, 3: GalA, 4: Glc, 5: Gal, 6: Ara.

**Figure 4 molecules-27-04759-f004:**
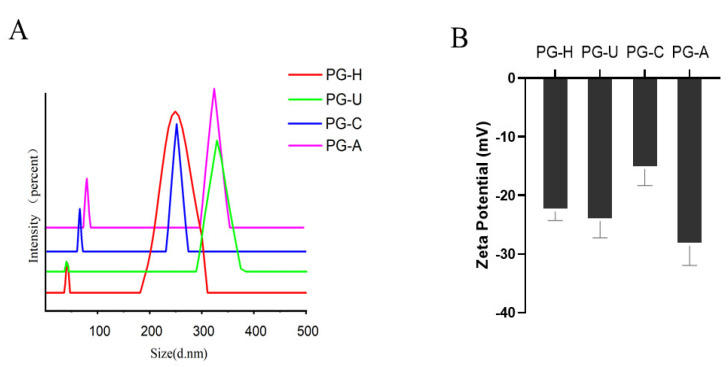
(**A**) Particle size distribution of PGs obtained by different extraction methods. (**B**) Zeta-potential of PGs obtained by different extraction methods.

**Figure 5 molecules-27-04759-f005:**
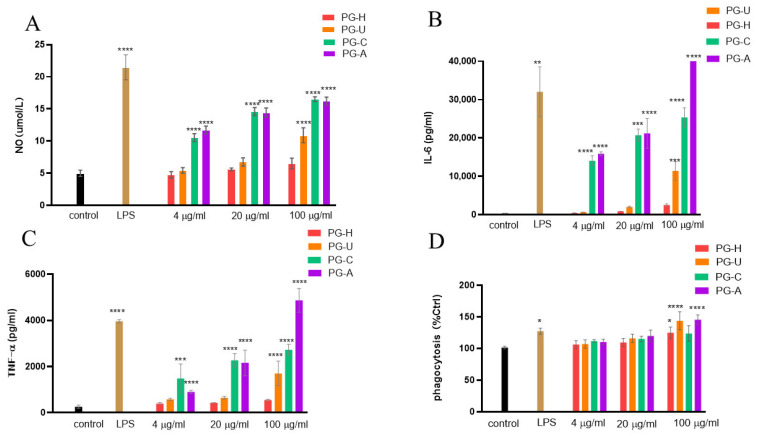
Effects of PGs on NO production (**A**), IL-6 production (**B**), TNF-α generation (**C**), an phagocytic activity (**D**) of RAW264.7 cells. * *p* < 0.05, ** *p* < 0.01, ****p* < 0.001, **** *p* < 0.001.

**Figure 6 molecules-27-04759-f006:**
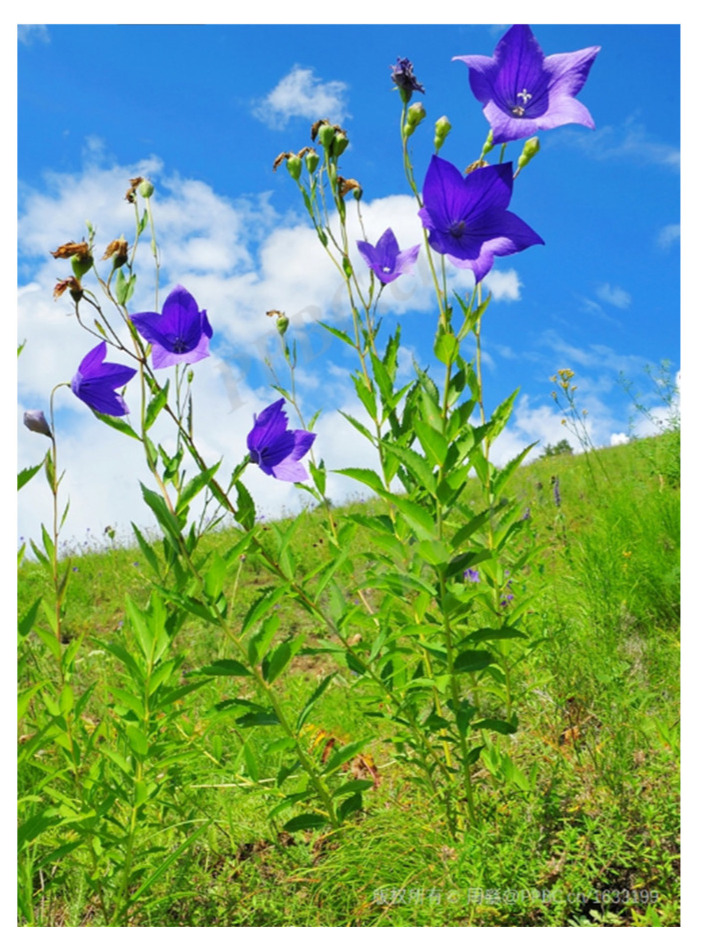
*Platycodon grandiflorum*.

**Table 1 molecules-27-04759-t001:** Effect of different extraction methods on the yield and carbohydrate content.

Samples	PG-H	PG-U	PG-C	PG-A
Yield (%)	16.6	7.1	3.8	2.8
Carbohydrate content (%)	80.9	94.0	94.5	92.3

**Table 2 molecules-27-04759-t002:** Molecular weight (kDa) distribution of PGs.

Sample	PG-H	PG-U	PG-C	PG-A
Mw of Peak I	>4414.3	>5141.5	>3925.5	>3479.0
Peak area ratio of Peak I (%)	42.5	33.7	53.4	71.7
Mw of Peak II	2.8	3.0	1.3	1.3
Peak area ratio of Peak II (%)	57.5	66.3	46.6	28.3

**Table 3 molecules-27-04759-t003:** Effects of different extracted methods on monosaccharide type and ratio.

	PG-H	PG-U	PG-C	PG-A
Man (mol %)	1.9	3.6	0.9	2.2
Rha (mol %)	2.7	3.2	4.2	7.2
GalA (mol %)	5.3	3.4	7.4	23.9
Glc (mol %)	22.8	42.2	5.8	11.2
Gal (mol %)	17.1	17.0	22.5	15.1
Ara (mol %)	50.1	30.6	59.2	40.4

## Data Availability

The datasets are available in article.

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
