# Peer review of "Comparative Characterization and Immunomodulatory Activities of Polysaccharides Extracted from the Radix of Platycodon grandiflorum with Different Extraction Methods"

_molecules, 2022, doi:10.3390/molecules27154759_

Round 1
Reviewer 1 Report
1. Title: why the “Polysaccharide” is distinguished by the use of a capital letter?
2. Why the Authors use PGs as an abbreviation of “polysaccharide”? PS/PSs (plural) is an official abbreviation
3. In introduction is really needed information about where this “polysaccharide” is present exactly in the P. grandiflorum.
4. Abstract: please don’t use the all abbreviations in the abstract and the details about yields (show them in results section)
5. After first using Platycodon grandiflorum please later use P. grandiflorum
6. Introduction: (line 42) please replace “macrophages” for “cells”.
7. Line 55: “methods”
8. Line 67: Please remove all sentence (it means exactly the same as the earlier one)
9. If is it assumed that by alkaline and acid extractions the PG is hydrolyzed the comparison of all method is useless...
10. Line 73: PG-W???
11. Tables 1, 2: in % values please use the numbers with only one decimal place
12. Lines 94-95: “...corresponds to fraction of polysaccharides with high molecular weight”, “....fraction of polysaccharides with lower molecular weight...”
13. Line 95: remove information about shape of peak this is “narrow” – the separation of polysaccharide mixture has given the two main peaks, but we can see that in all fraction the first peak consists two main different population of polysaccharides. Also the second peak of PG-C consists the two different polysaccharide populations.
14. Line 114: Glucose it is Glc
15. Line 115: molar ratio of what?
16. Line 119: Every time the polysaccharides were a samples...TFA hydrolysis should not be different in each sample
17. Lines 99-101: please correct
18. Molecular weight: The Reviewer would be careful with description like: “ the molar weights of PG-C and PG-A were significantly smaller” or “.... the stability of PG-A was best, followed by ...” (explanation is needed), Please insert the dextran standards separation to comparison in Fig. 2.
19. What is the molecular weight of PG-U? It is really important information, especially for explanation of further experiments.
20. Line 103: What does mean the sentence: “However, ....”???
21. Line 138: GalA
22. Line 146: ...on macrophage cells proliferation
23. Line 182: important note: all four polysaccharides have got the same monosaccharide composition. The difference between them is in molecular weights, different proportion of reducing ends and non-reducing ends, various properties...but still the monosaccharide composition is the same to the time the same monosaccharides are present. Please remember about this even in Conclusion section
24. Method about monosaccharide composition: How the Authors know about monosaccharide composition if the standards were started after “determination of the monosaccharide composition”? By which method the composition was determined?
25. Line 184: Molecular weight of 3500kDa is definitely not a small mass for polysaccharide.
26. Figure 6: LPS not lps
27. Line 230: sample rather than residue
28. pH 3.0 and other (please don’t use “pH=”
29. Line 232: “the same number of herbs”???
30. Line 235: in case the supernatants extracted by NaOH (pH 12) were neutral...” what was the pH value of supernatants extracted by water in different conditions? (PG-H and PG-U)?
31. Line 241: please remove “to a constant weight”
32. Line 242: rather “deproteined”
33. Line 243: rather sample then “residue”
34. The pattern for extraction yield is not needed.
35. Line 255: “phenol-sulfuric”
36. Please insert more details about FT-IR and UV analyses (description of samples preparation)
37. Line 277: “kDa”
38. Line 344: space
39. Line 350: “...the PGs extracted by different methods”
Author Response
Dear reviewer:
Thank your suggestions . We have provied response in the attachment.
Yours sincerely,
Wanwan xiao
Emaile:W989700@163.com.

Reviewer 2 Report
In the present manuscript, the effects of four different extraction methods, including hot water extraction (HW), ultra-sonic-assisted extraction (UAE), acid-assisted extraction (CAE), and alkali-assisted extraction (AAE) on the polysaccharide from Platycodon grandiflorum radix were evaluated.
Lines 3, 10: correct to "polysaccharides".
The abbreviation of PG for polysaccharides is misleading because it refers to Platycodon grandiflorum. It is better to revise to PS.
Introduction has to be extended.
Line 73: PG-W is not listed in Table 1 or referred in the abstract.
The aim of this study is confusing. The authors state in the title and the abstract that their was to extract and chemically characterize polysaccharides and evaluate their immunomodulatry activities. However, for the presented results its seems that the authors extracted monosaccharides. Morever, since the authors did not isolate specific compounds it is not clear how the observed bioactive properties where associated with specific compounds and polysaccharides in particular.
Author Response
Dear editor:
Thank your suggestions. We have proviede responses in the attachment.
Yours sincerely
Wanwan xiao
Emaile:W989700@163.com.

Reviewer 3 Report
Authors should explain abbreviations used in the Abstract - line 14
......polysaccharide from Platycodon grandiflorum (PG-H, PG-U, PG-C, PG-A).
Also in the text polysaccharides that were assigned as above should be described (e.g chemical structure).
It would be nice to have a photo of the plant in M&M section.
Author Response

(The authors gave the same response as above.)

Round 2
Reviewer 2 Report
The authors revised the manuscript taking into account my comments. Therefore, I recommend the acceptance of the manuscript in its present form.
Reviewer 3 Report
Changes in the text are made in accordance to the reviewers comments.